# CONTINUOUS HAND GESTURE SPOTTING THROUGH DEEP SEQUENTIAL ENCODING AND PROBABILISTIC TIME-SERIES MODELING

## ABSTRACT

Continuous hand gesture spotting in real time is a challenging problem because ambiguous gesture boundaries and abundant non-gesture motions often confound recognition systems. Unlike isolated recognition, spotting requires detecting both the onset and offset of gestures while rejecting irrelevant transitions, making robustness crucial for practical human–computer interaction. We present a hybrid framework that integrates MediaPipe Hands for extracting 3D landmarks, an LSTM Autoencoder for compact spatiotemporal encoding, and Gaussian Hidden Markov Models (HMMs) for probabilistic sequence modeling. To further suppress spurious detections during transitions, we introduce an ergodic threshold mechanism that adaptively filters low-likelihood segments. On a vocabulary of 10 command gestures, the system achieves 96.56% recognition accuracy, 97.89% segmental F1, and 6.55% word error rate (WER) in continuous input streams, while remaining lightweight enough to run on a CPU-only device. These results show that combining deep representation learning with probabilistic dynamics yields reliable boundary detection without heavy computational overhead. Beyond empirical gains, the framework is data-efficient and readily extensible to new vocabularies, enabling rapid adaptation with limited training data. Overall, these findings demonstrate the practical feasibility of robust gesture spotting, bridging the gap between controlled research settings and real-world applications in VR/AR environments and customizable user interfaces.

## 1 INTRODUCTION

Gesture recognition has become a core area in human–computer interaction (HCI), enabling natural and intuitive communication between humans and machines Khan & Ibraheem (2012). Applications span immersive VR/AR systems Lei et al. (2023), smart home control Dewangga et al. (2024), and sign language interpretation Koller et al. (2018). Despite this progress, continuous gesture spotting remains a challenging problem due to spatiotemporal variability, ambiguous gesture boundaries, and the frequent occurrence of non-gestural motions in natural hand movement streams Lee & Kim (1999).

Recent surveys Sarowar et al. (2025); Linardakis et al. (2025) emphasize that while isolated gesture recognition has matured, continuous spotting remains an open challenge because of incidental hand motions, uncertain gesture boundaries, and the need for real-time, low-latency inference. New benchmarks such as IPN Hand and IPN HandS Benitez-Garcia et al. (2025) highlight the prevalence of non-gesture segments in natural recordings, further underscoring the importance of robust rejection mechanisms.

To address these challenges, we propose a hybrid framework that integrates MediaPipe Hands for robust 3D hand tracking Lugaresi et al. (2019); Zhang et al. (2020), an LSTM Autoencoder for compact spatiotemporal feature encoding Malhotra et al. (2016), and Gaussian HMMs for probabilistic sequence modeling Zheng et al. (2021); Gruhl & Sick (2016), which extend standard HMMs to handle vector-valued inputs Huang et al. (1990); Rabiner (1989). To further suppress false detections, an ergodic threshold model Lee & Kim (1999) is incorporated to reject non-gesture segments.

This combination provides a principled approach for continuous gesture recognition by uniting deep feature learning with sequential probabilistic dynamics.

The key contributions of this work are threefold. First, the integration of MediaPipe Hands and an LSTM Autoencoder facilitates compact and stable spatiotemporal feature extraction. Second, an ergodic HMM–based threshold model provides a reliable way to discriminate gestures from non-gestures. Third, empirical validation using 10 command gestures shows high accuracy for both isolated recognition and continuous spotting tasks.

## 2  RELATED WORK

Probabilistic and deep learning approaches have long been applied to gesture recognition. Hidden Markov Models (HMMs) remain a classical choice due to their ability to capture temporal structures, but they typically rely on handcrafted or simplified features, which limits robustness in complex real-world settings Lee & Kim (1999); Rabiner (1989). Conditional Random Fields (CRFs) have also been investigated for sequence labeling, providing stronger boundary modeling but at the expense of high computational cost Wang et al. (2006).

With the advent of deep learning, CNN–RNN hybrids and LSTM-based Autoencoders have demonstrated strong representation learning capabilities, particularly for pre-segmented gestures Malhotra et al. (2016). More recently, Transformer-based models have achieved state-of-the-art performance in continuous sign spotting by capturing long-range dependencies Camgoz et al. (2020); Varol et al. (2021). In addition, multimodal fusion that combines vision with wearable sensors, event cameras, or electromyography (EMG) signals has been explored to enhance robustness under occlusion and viewpoint changes Lei et al. (2023).

However, most deep learning methods still assume isolated or pre-segmented inputs, limiting their effectiveness in continuous gesture spotting where gesture boundaries are ambiguous Malhotra et al. (2016); Camgoz et al. (2020); Varol et al. (2021). Threshold-based rejection models have been introduced to filter non-gesture segments, yet conventional designs are often too simplistic to adapt to diverse spatiotemporal variations Lee & Kim (1999).

Recent continuous gesture frameworks, such as OO-dMVMT Cunico et al. (2023), explicitly address latency and segmentation accuracy in skeleton-based settings. Gesture Spotter Shen et al. (2022) instead emphasizes single-time activation and activation lag in VR/AR contexts. However, these approaches generally lack adaptive thresholding to reject transitional non-gesture segments.

Our work addresses these limitations by integrating MediaPipe Hands for robust 3D hand landmark tracking Lugaresi et al. (2019); Zhang et al. (2020), an LSTM Autoencoder for compact spatiotemporal feature extraction Malhotra et al. (2016), Gaussian HMMs for sequential probabilistic modeling Zheng et al. (2021); Rabiner (1989), and an ergodic threshold model for adaptive non-gesture rejection Lee & Kim (1999). This hybrid framework combines the structural modeling strengths of HMMs with the representational power of deep learning, explicitly targeting the challenge of spotting gestures from continuous, unsegmented motion streams.

## 3  METHODOLOGY

Our framework integrates MediaPipe Hands, an LSTM Autoencoder, Gaussian HMMs, and a threshold-based Gesture Spotting Network (GSN) to achieve robust continuous gesture recognition.

### 3.1  MEDIAPIPE HANDS

MediaPipe Hands, a real-time hand-tracking framework developed by Google, extracts 21 normalized 3D landmarks $(x, y, z)$ from a single RGB image Lugaresi et al. (2019); Zhang et al. (2020). Its two-stage pipeline consists of palm detection using BlazePalm and regression-based landmark estimation. Sequenced over time, these landmarks form gesture trajectories. To ensure consistency across samples, preprocessing includes center alignment, scale normalization, and low-pass filtering.

## 3.2 LSTM AUTOENCODER

To capture spatiotemporal dependencies, we employ a multi-layer LSTM Autoencoder Malhotra et al. (2016). The encoder compresses landmark sequences into a latent vector, while the decoder reconstructs the original sequence. The model is trained in an unsupervised manner by minimizing mean squared error (MSE), ensuring that the latent representation captures compact and discriminative spatiotemporal patterns. After training, the encoder's latent vectors serve as feature representations for gesture modeling with HMMs.

## 3.3 GAUSSIAN HMM

Each gesture class is modeled by an independent Gaussian HMM Zheng et al. (2021); Gruhl & Sick (2016); Huang et al. (1990); Rabiner (1989), where emission probabilities follow multivariate Gaussian distributions.

A HMM is defined by four components:

- $Q = \{q_1, q_2, \ldots, q_N\}$: the set of hidden states,
- $\pi = \{\pi_i\}$: the initial state distribution, where $\pi_i = P(s_1 = q_i)$,
- $A = \{a_{ij}\}$: the state transition probabilities, $a_{ij} = P(s_{t+1} = q_j \mid s_t = q_i)$, and
- $B = \{b_i(x)\}$: the observation probability distributions, modeled here as Gaussians with $b_i(x) = \mathcal{N}(x; \mu_i, \Sigma_i)$.

For an observation sequence $X = (x_1, x_2, \ldots, x_T)$, the joint probability sequence $S$ is

$$P(X, S) = \pi_{s_1} \cdot b_{s_1}(x_1) \cdot \prod_{t=2}^{T} a_{s_{t-1}, s_t} \cdot b_{s_t}(x_t) \,.$$

For each gesture class $G_k$ an independent HMM $\lambda_k = (\pi_k, A_k, B_k)$ is trained, with parameters estimated using the Baum–Welch algorithm. The number of states $N$, training iterations, and convergence criteria were tuned through preliminary experiments.

At test time, we evaluate the log-likelihood $\log P(X \mid \lambda_k)$ of the input sequence under each model, and select the most likely gesture class:

$$\hat{k} = \arg\max_k \log P(X \mid \lambda_k) \,.$$

The log-likelihood is computed with the Forward algorithm, which emphasizes spotting performance rather than exact state sequence recovery. In addition to classification, Gaussian HMMs support boundary detection in continuous input streams through sliding-window log-likelihood evaluation.

## 3.4 THRESHOLD MODEL AND GESTURE SPOTTING NETWORK

Continuous sequences inevitably include transitional or non-gesture motions that may trigger false detections. To address this, we introduce a threshold model Lee & Kim (1999), defined by a class-specific threshold:

$$\theta_k = \mu_k - \alpha \cdot \sigma_k$$

where $\mu_k$ and $\sigma_k$ denote the mean and standard deviation of training log-likelihoods, and $\alpha$ (set to 1.0) is a confidence constant. Implemented as an ergodic HMM, the threshold model adapts to diverse sub-patterns and suppresses false detections. In continuous spotting, gesture likelihoods are evaluated over sliding windows:

$$\ell_t^k = \log p(x_{t-w:t} \mid \lambda_k) \,.$$

A segment is accepted as a gesture $k$ only if $\ell_t^k \geq \theta_k$; otherwise, it is classified as a non-gesture.

As shown in Figure 1, gesture-specific Gaussian HMMs produce log-likelihood scores that are compared against the adaptive threshold $\theta_k$. Segments falling below this boundary are classified as

non-gestures, effectively suppressing false detections caused by transitional or irrelevant motions. This mechanism improves boundary localization in continuous streams by filtering out spurious responses while retaining valid gesture intervals.

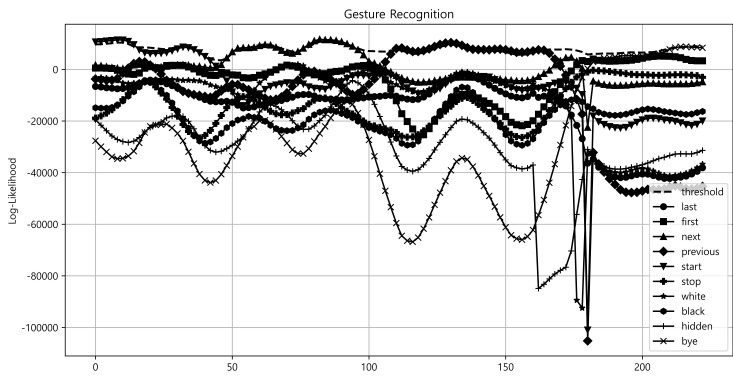

Figure 1: Log-likelihood comparison between gesture models and the ergodic threshold model. The threshold suppresses low-likelihood segments corresponding to transitional or non-gesture motions.

All gesture HMMs and threshold models are then combined into a unified GSN. As illustrated in Figure 2, each gesture class is represented by a dedicated Gaussian HMM, while a corresponding ergodic threshold model captures non-gesture patterns. The GSN continuously evaluates input sequences from a virtual initial state, computing log-likelihoods in parallel across all models and dynamically selecting the most probable hypothesis. This architecture reduces false positives, handles ambiguous gesture boundaries, and scales efficiently to new gesture vocabularies, enabling robust gesture spotting in unsegmented input streams.

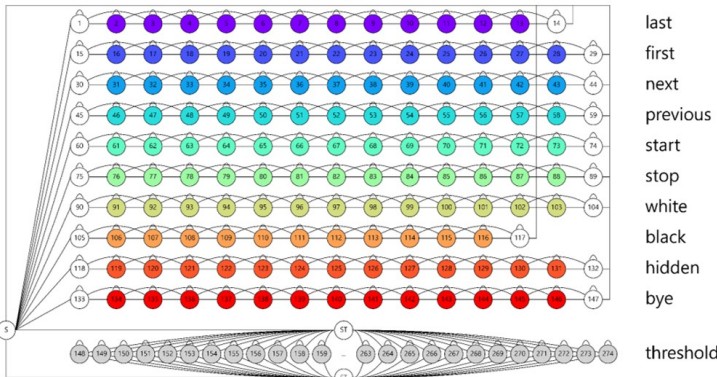

Figure 2: Structure of the GSN. Gesture-specific HMMs and threshold models are integrated within a unified framework, enabling continuous evaluation of input streams and robust gesture spotting.

# 4 EXPERIMENTAL SETUP AND RESULTS

## 4.1 EXPERIMENTAL SETUP

We evaluated our framework on a vocabulary of 10 command gestures (*next*, *previous*, *start*, *stop*, *white*, *black*, *hidden*, *bye*, *last*, *first*) designed for Microsoft PowerPoint control, using only the right hand for consistency. For each gesture, 150 samples were collected (100 for training and 50 for isolated testing), yielding 1,500 sequences with 15–72 frames each.

For continuous spotting, a separate test set of 170 sequences was constructed, each containing 3–4 gestures (553 gestures in total). MediaPipe Hands extracted 21 3D landmarks per frame, followed by center alignment, scale normalization, and low-pass filtering. An LSTM Autoencoder (latent dimension 256) was trained with the Adam optimizer (learning rate 0.001) to minimize MSE, and

encoder latent vectors were used as features for Gaussian HMMs (13–15 states, diagonal covariance). Threshold models were implemented as ergodic HMMs.

All gesture and threshold models were integrated into the GSN. Experiments were conducted on a CPU-only laptop (Intel i7, 32 GB RAM) to demonstrate deployment feasibility without specialized hardware.

## 4.2 EVALUATION PROTOCOL

Performance was evaluated in two stages. First, isolated gesture recognition was assessed by testing individual gestures to validate accuracy and confirm that the threshold models operated properly. The goal was not to maximize accuracy on large test sets but to ensure that each gesture consistently exceeded a satisfactory recognition level (set at 90% in this study). Second, continuous gesture spotting was examined, comprising both segmentation and recognition. Boundary detection performance was measured using the Segmental F1-score, while recognition accuracy was evaluated with overall correctness and Word Error Rate (WER), which jointly capture insertion, deletion, and substitution errors.

The WER was defined as

$$\text{WER} = \frac{S + D + I}{N}$$

where $S$, $D$, and $I$ denote the numbers of substitution, deletion, and insertion errors, respectively, and $N$ is the number of ground-truth gestures. The Segmental F1-score was computed as

$$F1 = \frac{2 \cdot \text{Precision} \cdot \text{Recall}}{\text{Precision} + \text{Recall}}$$

where

$$\text{Precision} = \frac{TP}{TP + FP}, \quad \text{Recall} = \frac{TP}{TP + FN}.$$

Here, $TP$ (true positives) are correctly detected gestures, $FP$ (false positives) are non-gestures or incorrect detections misclassified as gestures, and $FN$ (false negatives) are missed gestures.

## 4.3 RESULTS OVERVIEW

Isolated gesture recognition rates are shown in Table 1. Each gesture was tested on 50 held-out samples, and recognition performance was averaged across 500 independent trials to account for stochastic variations in HMM initialization and training. The system achieved an average accuracy of 97.80%, even for challenging pairs such as white and hidden, which share visually similar initial trajectories. These results confirm that the proposed feature representation provides sufficient discriminability to support the construction of reliable threshold models and the GSN.

Table 1: Recognition rates for isolated gestures. Each gesture was tested with 50 samples, and the results were averaged over 500 trials.

| Gesture | Test Data | Correct | Recognition Rate(%) |
|---|---|---|---|
| last | 50 | 48 | 96.00 |
| first | 50 | 50 | 100.00 |
| next | 50 | 50 | 100.00 |
| previous | 50 | 49 | 98.00 |
| start | 50 | 50 | 100.00 |
| stop | 50 | 50 | 100.00 |
| white | 50 | 48 | 96.00 |
| black | 50 | 49 | 98.00 |
| hidden | 50 | 49 | 98.00 |
| bye | 50 | 47 | 94.00 |
| **Sum** | **500** | **489** | **97.80** |

The evaluation of the threshold model is presented in Table 2 using the Segmental F1-score, which jointly considers precision and recall at the segmentation level. In this calculation, correctly localized gestures are counted as True Positives. Substitution errors are treated both as False Positives (incorrectly inserted gestures) and False Negatives (missed ground-truth gestures), while deletion errors are treated as False Negatives. With this evaluation protocol, the system achieved a 97.89% segmental F1-score, demonstrating robust boundary detection even in the presence of transitional hand motions.

Table 2: Threshold model evaluation using the Segmental F1-score. True Positives denote correctly segmented gestures; substitution errors are counted as both False Positives and False Negatives, while deletion errors are counted as False Negatives.

| Gesture | True Positive | False Positive | False Negative | Precision(%) | Recall(%) | Segmental F1-score(%) |
|---|---|---|---|---|---|---|
| last | 49 | 0 | 4 | 100.00 | 92.45 | 96.08 |
| first | 49 | 0 | 2 | 100.00 | 96.08 | 98.00 |
| next | 49 | 1 | 1 | 98.00 | 98.00 | 98.00 |
| previous | 57 | 1 | 0 | 98.28 | 100.00 | 99.13 |
| start | 60 | 0 | 2 | 100.00 | 96.77 | 98.36 |
| stop | 58 | 1 | 1 | 98.31 | 98.31 | 98.31 |
| white | 55 | 1 | 0 | 98.21 | 100.00 | 99.10 |
| black | 53 | 0 | 5 | 100.00 | 91.38 | 95.50 |
| hidden | 50 | 0 | 0 | 100.00 | 100.00 | 100.00 |
| bye | 54 | 0 | 4 | 100.00 | 93.10 | 96.43 |
| **Total** | **534** | **4** | **19** | **99.26** | **96.56** | **97.89** |

Table 3 presents results on continuous gesture spotting across 170 test sequences containing 553 gestures in total. The system achieved a recognition accuracy of 96.56% and a WER of 6.55%, where WER accounts for insertion, deletion, and substitution errors at the sequence level. These findings confirm that integrating deep sequential encoding with probabilistic modeling yields reliable performance in realistic continuous input streams, validating the proposed framework for practical HCI applications.

Table 3: Continuous gesture spotting results across 170 test sequences. WER includes insertion, deletion, and substitution errors.

| Gesture | Test Data | Insertion Error | Deletion Error | Substitution Error | Correct | Recognition Rate(%) | WER(%) |
|---|---|---|---|---|---|---|---|
| last | 53 | 1 | 3 | 1 | 49 | 92.45 | 10.20 |
| first | 51 | 3 | 1 | 1 | 49 | 96.08 | 10.20 |
| next | 50 | 2 | 0 | 1 | 49 | 98.00 | 6.12 |
| previous | 57 | 2 | 0 | 0 | 57 | 100.00 | 3.51 |
| start | 62 | 1 | 2 | 0 | 62 | 96.77 | 5.00 |
| stop | 59 | 1 | 1 | 0 | 58 | 98.31 | 5.17 |
| white | 55 | 4 | 0 | 0 | 55 | 100.00 | 7.27 |
| black | 58 | 0 | 4 | 1 | 53 | 91.38 | 9.43 |
| hidden | 50 | 0 | 0 | 0 | 50 | 100.00 | 0.00 |
| bye | 58 | 1 | 4 | 0 | 54 | 93.10 | 9.26 |
| **Sum** | **553** | **16** | **15** | **4** | **524** | **96.56** | **6.55** |

Overall, the results demonstrate strong performance in both isolated and continuous settings. Remaining errors primarily occur in short or visually similar gestures, where temporal overlap reduces separability. These cases are analyzed further in the subsequent Error Analysis and Discussion.

# 5 ERROR ANALYSIS AND DISCUSSION

## 5.1 GESTURE DISCRIMINATION

Recognition errors primarily occurred in short gestures (15–30 frames) and in gestures with similar initial movements. For example, *white* (lightly clenching the fist near the nose) and *hidden* (clenching followed by finger extension) begin almost identically, reducing separability and lowering likelihood scores. Similar challenges with short-duration or fine-motor gestures have been noted in prior sequential recognition studies Khan & Ibraheem (2012); Koller et al. (2018), underscoring the importance of sufficient temporal context and the design of distinct gesture vocabularies.

## 5.2 GESTURE SPOTTING

Spotting errors were categorized into insertion, deletion, and substitution. Insertion errors occurred when transitional or non-gestural motions were misclassified as gestures, fragmenting true sequences. Deletion errors occurred when consecutive gestures merged, causing partial recognition loss. Substitution errors reflected confusion between gestures with overlapping motion patterns.

In spotting tasks, insertion errors are especially critical because they often stem from limitations of the threshold model.

Gestures such as *last*, *black*, and *bye* exhibited relatively high deletion error rates and correspondingly lower Segmental F1-scores. These gestures are characterized by motion patterns where finger configurations remain constant while only hand position changes. Deletion errors arose when the gesture was entirely missed or when its short duration caused preparatory motion and the actual gesture to merge.

The gesture *first* exhibited a different error pattern. Like *last*, it involves index and middle finger movements with horizontal hand motion, but it also contains repeated finger actions. Since most other gestures exhibit little finger dynamics, the LSTM Autoencoder emphasized translational motion over finger articulation. Consequently, horizontal movements with slight finger activity were frequently misrecognized as *first*, leading to insertion errors. Even simple gestures such as *white* occasionally incurred insertion errors when transitional hand motions were misclassified as valid gestures Koller et al. (2018); Lee & Kim (1999).

## 5.3 DISCUSSION

Gesture similarity and short temporal duration were the primary sources of recognition errors. The reliance on log-likelihood thresholds further amplified these difficulties, as subtle overlaps between gesture onset and transitional motions reduced class separability. Although the ergodic threshold model effectively filtered many non-gesture segments, borderline cases still persisted and led to spotting errors.

These findings point to several promising directions for improvement. First, temporal modeling with longer context windows or hybrid LSTM-HMM structures could better capture long-range dependencies, thereby improving recognition of short gestures. Second, discriminative training objectives such as Maximum Mutual Information (MMI) or Connectionist Temporal Classification (CTC) can explicitly penalize confusions between visually or temporally similar gestures. Finally, multimodal integration may further enhance discriminability for subtle or ambiguous gestures. This approach combines visual landmarks with inertial, depth, or EMG signals. These findings point to several promising directions for improvement.

At the same time, the results highlight the framework's practical advantages. Because new gesture models can be trained with limited data, the system can adapt rapidly to redefined or personalized gesture vocabularies. This flexibility makes the approach particularly suitable for VR/AR environments and user-defined gesture interfaces, where scalability and adaptability are essential for deployment. Nevertheless, the current design reflects a trade-off between simplicity and accuracy: the threshold mechanism remains lightweight and efficient but does not fully resolve borderline gesture cases, motivating the more advanced strategies outlined above.

## 6 CONCLUSION AND FUTURE WORK

This work presented a hybrid framework for continuous gesture recognition that integrates MediaPipe Hands, an LSTM Autoencoder, Gaussian HMMs, and a threshold-based GSN. Evaluations on a vocabulary of 10 command gestures demonstrated 96.56% recognition accuracy, 97.89% segmental F1-score, and a 6.55% WER, showing that combining deep sequential encoding with probabilistic modeling yields robust boundary detection in realistic HCI scenarios.

While the threshold model effectively reduces false detections, its scalability is limited: state complexity grows with the number of gesture classes, which can degrade spotting performance. Addressing this trade-off between simplicity and accuracy will be critical for broader deployment.

At the same time, the framework demonstrates strong adaptability. Because new gesture models can be trained from limited data, it is well suited for VR/AR applications and user-defined gesture interfaces, where flexibility and personalization are essential.

Future work will proceed along three directions. First, lightweight modeling aimed at designing thresholding mechanisms whose complexity is independent of the gesture vocabulary size. Second, dataset expansion to cover larger vocabularies and more diverse user groups, thereby improving robustness and generalization. Third, extension of the framework to broader application scenarios, extending beyond PowerPoint control to VR/AR interaction, sign language recognition, smart environments, and user-customized gesture interfaces.

## ETHICS STATEMENT

The gesture data used in this study were collected from voluntary adult participants with informed consent. No personally identifiable information was recorded, and all procedures followed institutional ethical guidelines. The framework is designed to enhance human–computer interaction in VR/AR environments. Potential risks such as misuse for surveillance or privacy intrusion were considered, and care was taken to minimize such concerns.

## USE OF LARGE LANGUAGE MODELS

Large Language Models (LLMs), such as ChatGPT, were used only to polish the English writing of this manuscript. All research ideas, methods, analyses, and results are original contributions of the authors.

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
