# OpenReview forum: "Continuous Hand Gesture Spotting through Deep Sequential Encoding and Probabilistic Time-Series Modeling"
_ICLR.cc/2026/Conference — ICLR 2026 Conference Withdrawn Submission_

### Official Review · Reviewer_HVaK · 2025-10-29

**Soundness:** 2
**Presentation:** 1
**Contribution:** 2
**Rating:** 2
**Confidence:** 3

**Summary:**

This work presents a hybrid framework that integrates MediaPipe Hands, an LSTM Autoencoder, Gaussian Hidden Markov Models (HMMs), and an ergodic threshold mechanism for continuous hand gesture spotting. Specifically, MediaPipe Hands is used to extract 3D landmarks, the LSTM Autoencoder performs compact spatiotemporal encoding, Gaussian HMMs are employed for probabilistic sequence modeling, and the ergodic threshold mechanism adaptively filters low-likelihood segments. The effectiveness of the proposed method is validated through experiments.

**Strengths:**

The proposed method is lightweight and can be deployed on CPU-only devices for continuous hand gesture spotting.

**Weaknesses:**

The paper shows very limited originality, as it merely combines several existing methods without introducing substantial innovation.

The experimental validation is weak, being conducted only on a simple dataset without any comparisons to existing approaches.

There are writing issues, and the citation style is clearly incorrect.

**Questions:**

How does the proposed method compare with the current state-of-the-art approaches for continuous hand gesture spotting?

What are the speed and resource consumption characteristics of the proposed method?

---

### Official Review · Reviewer_n6VW · 2025-11-04

**Soundness:** 2
**Presentation:** 2
**Contribution:** 1
**Rating:** 2
**Confidence:** 5

**Summary:**

The paper focuses on detecting gestures in continuous movements by identifying the onset of these gestures from their boundary and offset windows. It uses MediaPipe for pose estimation, then employs LSTM and graphical models for spatio-temporal modelling. The model then uses gesture-specific transitional scores and models to localize their onsets.

**Strengths:**

- The paper explicitly tackles continuous gesture spotting (onset/offset detection plus non-gesture rejection), which is more challenging and practically relevant than isolated gesture classification for HCI.

- The system is designed to run in real time on a CPU-only machine, which is useful for many HCI scenarios where GPU resources are not available.

**Weaknesses:**

- The temporal modelling relies on LSTM autoencoders and Gaussian HMMs with manually designed thresholding. While these choices are defensible from an efficiency standpoint, the field has largely moved toward attention-based and transformer models for continuous sign and gesture recognition. The paper cites such works but does not provide empirical comparisons, nor does it compare against simpler yet strong baselines. Without these baselines, it is challenging to justify the added complexity of the LSTM-encoder + HMM + threshold architecture or to support the claim that this hybrid design is indeed necessary.

- The experimental setup is restricted to 10 command gestures for PowerPoint control, using only the right hand. This is a narrow, application-specific vocabulary, and many gestures are relatively simple. The paper does not evaluate its approach on standard continuous-hand benchmarks (e.g., the datasets mentioned in the related work), so it is unclear how the approach would fare on more diverse, noisy, or complex gesture sets.

- The paper claims that the framework is “readily extensible to new vocabularies” and that it is data-efficient. However, there is no experiment where new gestures are added to the vocabulary to measure how performance scales with vocabulary size.

- Given that the threshold model complexity grows with the number of gesture classes, the scalability claim is particularly important to validate experimentally.

**Questions:**

I do not have questions for the authors.

---

### Official Review · Reviewer_XrpH · 2025-11-05

**Soundness:** 2
**Presentation:** 3
**Contribution:** 1
**Rating:** 2
**Confidence:** 3

**Summary:**

The paper presents a new integrated method for gesture spotting, i.e., the segmentation and recognition of gestures from a continuous stream of images. To this end, four methods are combined: "MediaPipe Hands" for localization, an LSTM autoencoder, a HMM for recognition, and an "ergodic" threshold mechanism to suppress spurious detections. This system is validated on a self-created dataset of continuous hand gestures with very good accuracy.

**Strengths:**

The paper is easy to read and describes all parts of the proposed system well. Results are rather credible, and the fact that this runs without GPU acceleration is pretty amazing.

**Weaknesses:**

- It would be better to introduce what "spotting" means, exactly.
- The related work section could be more extensive about "spotting", not just gesture recognition
- The novelty is rather incremental, as you "just" piece together elements that have been described already
- It would be better to test you method on publicly available datasets to allow a comparison to other methods
- The dataset you test you method on is not public, making the claims hard to verify
- The path from raw data to system response is not quite detailed enough (feature encoding, cropping, rescaling, ...). This should at least be explained in the appendix.

**Questions:**

- Do you assume that all gestures have the same length? If so, please explain why this is not a problem in practice
- Is the dataset you use for evaluation publicly available? I found no link or similar...

---

### Note · Authors · 2025-11-12

**Comment:**

In this work, I attempted to extend my earlier HMM-only approach to the ‘gesture spotting’ task by integrating it with deep learning and enabling vector-level inputs - rather than scalar inputs - so that fine-grained finger-level motion could be incorporated. This was intended to address the inefficiencies of deep learning when handling sequence data. However, due to time constraints, I was unable to complete the necessary experiments, and therefore I am withdrawing the submission.

**Withdrawal Confirmation:**

I have read and agree with the venue's withdrawal policy on behalf of myself and my co-authors.